# 50 Hz Magnetic Field Exposure Inhibited Spontaneous Movement of Zebrafish Larvae through ROS-Mediated syn2a Expression

**DOI:** 10.3390/ijms24087576

**Published:** 2023-04-20

**Authors:** Yixin Guo, Yiti Fu, Wenjun Sun

**Affiliations:** 1Bioelectromagnetics Key Laboratory, Zhejiang University School of Medicine, Hangzhou 310058, China; 2Institute of Environmental Medicine, Zhejiang University School of Medicine, Hangzhou 310058, China

**Keywords:** 50 Hz magnetic-field (MF), zebrafish, spontaneous movement (SM), window effect, synapsin IIa (syn2a), ROS

## Abstract

Extremely low frequency electromagnetic field (ELF-EMF) exists widely in public and occupational environments. However, its potential adverse effects and the underlying mechanism on nervous system, especially behavior are still poorly understood. In this study, zebrafish embryos (including a transfected synapsin IIa (syn2a) overexpression plasmid) at 3 h post-fertilization (hpf) were exposed to a 50-Hz magnetic field (MF) with a series of intensities (100, 200, 400 and 800 μT, respectively) for 1 h or 24 h every day for 5 days. Results showed that, although MF exposure did not affect the basic development parameters including hatching rate, mortality and malformation rate, yet MF at 200 μT could significantly induce spontaneous movement (SM) hypoactivity in zebrafish larvae. Histological examination presented morphological abnormalities of the brain such as condensed cell nucleus and cytoplasm, increased intercellular space. Moreover, exposure to MF at 200 μT inhibited syn2a transcription and expression, and increased reactive oxygen species (ROS) level as well. Overexpression of syn2a could effectively rescue MF-induced SM hypoactivity in zebrafish. Pretreatment with N-acetyl-L-cysteine (NAC) could not only recover syn2a protein expression which was weakened by MF exposure, but also abolish MF-induced SM hypoactivity. However, syn2a overexpression did not affect MF-increased ROS. Taken together, the findings suggested that exposure to a 50-Hz MF inhibited spontaneous movement of zebrafish larvae via ROS-mediated syn2a expression in a nonlinear manner.

## 1. Introduction

Extremely low frequency electromagnetic fields (ELF-EMFs), which frequency ranges from 0 to 300 Hz, are generated widely by electricity generation, transmission, and kinds of domestic and industrial electric devices. Since Wertheimer and Leeper first reported that 60-Hz MF exposure, which was produced by high-current configurations, was associated with increased incidence of childhood leukemia in 1979 [1], the potential adverse effects of ELF-EMF on human health has aroused public concern. Studies over the past four decades have identified the nervous system as a target organ of ELF-EMF. Epidemiological studies found that there existed close associations between ELF-EMF exposure and diseases of the nervous system such as brain tumor [2], Alzheimer’s disease (AD) [3,4,5], amyotrophic lateral sclerosis (ALS) [6,7] and poor sleep quality [8]. Due to the fact that function impairment of the nervous system could consequently show behavioral changes, the behavior is generally recognized as an integrated sensitive indicator in neurotoxicity evaluation [9]. Studies on behavioral effects also showed that ELF-EMF exposure could induce anxiety, stress-like behaviors, depressed behaviors, impairment of learning, memory and cognitive function [10]. However, research on locomotor behavior has not reached a consensus. Some studies found that ELF-EMF exposure inhibited locomotor behavior [11,12,13], but some believed that ELF-EMF exposure promoted it [14,15,16], ever others considered that ELF-EMF has no effect on locomotor behavior [17,18,19,20,21,22,23].

On the other hand, the mechanism of behavioral abnormalities induced by ELF-EMF exposure has also been explored. Özgün et al. [24] reported that N-methyl-D-aspartate (NMDA) receptor activation mediated 50-Hz MF-induced human neuronal differentiation. ELF-EMF exposure could decrease acetylcholinesterase (AChE) activity [25] and degeneration of the cholinergic pathways [26]. Other studies found that exposure to ELF-EMF could affect the reactivity of D-1 dopamine receptors [16,27], and potentiate morphine-induced decrease in D-2 dopamine receptor density [28]. In addition, an increase in free radicals such as reactive oxygen species (ROS) and nitric oxide (NO) in hypothalamus was involved in obsessive compulsive disorder- and anxiety-like behavior induced by ELF-EMF exposure [29,30,31]. A 50-Hz MF was even considered as a “stressor” that elevated proopiomelanocortin mRNA level and led to depressive-like behavior [32]. However, it is far from clarifying clearly the relationship between ELF-EMF and behavioral effects.

As a vertebrate animal model for investigating the developmental neurotoxicity, the zebrafish (*Danio rerio*) has received increased popularity in recent years [33,34]. Especially, the swimming behavior of zebrafish is the sum of activities that are controlled by the nervous system in response to external stimuli [35]. To determine the possible influence of ELF-EMF exposure on locomotor behavior, in the present study, the effects and related mechanisms of 50-Hz MF exposure on the swimming behavior were investigated and verified in zebrafish embryos and larvae.

## 2. Results

### 2.1. Exposure to a 50-Hz MF Did Not Affect the Basic Development on Zebrafish

To study the possible effects of 50-Hz MF exposure on development, wild-type zebrafish embryos at 3 h postfertilization (hpf) were exposed to a 50-Hz MF with a series of intensities (100, 200, 400 and 800 μT, respectively) for 1 h or 24 h every day for 5 days. The hatching rate at 72 hpf, mortality and malformation rate at 120 hpf were scored. However, results did not show any significant differences among treatment groups under current exposure conditions compared to the sham (*p* > 0.05) (Table 1). It means that 50-Hz MF exposure did not affect the basic development of zebrafish embryos/larvae.

### 2.2. Effects of 50-Hz MF Exposure on Locomotor Behavior

Locomotor behavior of zebrafish appears sequentially during development, which include an early, transient period of spontaneous tail coiling, followed by escape response to touch, and free swimming [36]. To determine the effect of 50-Hz MF exposure on the development of locomotor behavior, zebrafish embryos at 3 hpf were exposed to a 50-Hz MF with a series of intensities (100, 200, 400 and 800 μT, respectively) for 1 h or 24 h every day for 5 days, and three types of locomotion behavior: spontaneous tail coiling of embryos at 24 hpf, spontaneous movement (SM) at 120 hpf, and swimming in response to light-dark transition at 120 hpf were evaluated. Results showed that, in general, the frequency of spontaneous tail coiling of embryos at 24 hpf was diminished after exposure to a 50-Hz MF with different intensities and time, while there was no statistical significance compared with the sham group (*p* > 0.05) (Figure 1A). However, exposure zebrafish embryos to 50-Hz MF with 200 μT for either 1 h or 24 h everyday could significantly decrease movement distance and average speed at 120 hpf during 10 min SM, yet the other intensities of MF could not produce inhibitory effects on SM (Figure 1B) (Table 2). The results of light-dark transition test demonstrated a higher swimming speed in dark period followed by a decreased swimming speed after the light stimulus, which is a normal movement rhythm and periodic changes, and there was no statistical significance between different intensities of MF exposure (Figure 1C). Namely, exposure to a 50-Hz MF could significantly inhibit spontaneous movement but did not affect light-dark transition locomotor response in zebrafish larvae.

### 2.3. The Effects of 50-Hz MF Exposure on Brain Tissue Morphology and Neurodevelopment-Related Gene Expression in Zebrafish

In order to explore the mechanisms underlying MF-induced SM inhibition, zebrafish embryos were exposed to a 50-Hz MF with 200 μT for 1 h or 24 h every day for 5 days. The brain tissue morphology was observed by HE staining, and the expression of neurodevelopment-related genes at 120 hpf were analyzed by qRT-PCR and Western blot. The results of HE staining of the zebrafish brain revealed that neurons were arranged neatly and densely, the structure was intact and clear, the intercellular space was dense and without edema in sham group. However, the nucleus and cytoplasm of cells in the brain tissue was concentrated and the intercellular space was expanded after MF exposure (Figure 2A). Results of qRT-PCR showed that there was no significant alteration of *elavl3*, *gap43* and *shha* transcription, while *gfap* transcription was down-regulated by MF exposure for 24 h every day for 5 days, and *mbp* transcription was down-regulated only when MF exposure for 1 h every day for 5 days. Interestingly, the transcriptional level of *α1-tubulin* and *syn2a* (*synapsin IIa*) were down-regulated significantly in both MF exposure modes (Figure 2B), but only the syn2a protein expression was down-regulated significantly (Figure 2C), indicating that syn2a would be closely related to MF-induced SM hypoactivity in zebrafish.

### 2.4. Overexpression of syn2a Restored MF-Induced SM Hypoactivity in Zebrafish

In order to confirm the possibility that a reduced syn2a protein expression mediates MF-induced SM hypoactivity in zebrafish, pCS2^+^-syn2a-EGFP plasmid which could overexpress syn2a was microinjected into one-cell-stage zebrafish embryos before 1 hpf. The pCS2^+^ plasmid vector containing only EGFP (pCS2^+^-EGFP plasmid) served as a control. Results showed that both pCS2^+^-syn2a-EGFP and pCS2^+^-EGFP plasmids could express green fluorescence protein in zebrafish embryos at 24 hpf (Figure 3A) and pCS2^+^-syn2a-EGFP plasmid significantly overexpressed syn2a in zebrafish larvae at 120 hpf (*p* < 0.01) (Figure 3B), indicating that both pCS2^+^-EGFP and pCS2^+^-syn2a-EGFP plasmids worked well in zebrafish embryos and larvae.

The embryos transfected with plasmid vector or syn2a-overexpressed plasmid were cultured to 3 hpf, divided into groups randomly and exposed to a 50-Hz MF with 200 μT for 1 h or 24 h for 5 days. After removing the embryos without green fluorescence under blue excitation light at 24 hpf, the rest of embryos (Figure 3C) were cultured to 120 hpf for SM detection. The results showed that plasmid vector did not affect the MF-induced SM hypoactivity, while syn2a-overexpressed plasmid could mitigate MF-induced SM hypoactivity of zebrafish larvae (Figure 3D). It suggested that syn2a mediated SM hypoactivity caused by MF exposure.

### 2.5. MF-Induced SM Hypoactivity Depended on ROS

Previous studies demonstrated that ROS is a definite target molecule for MF interacting with biological system [37]. To explore the possible role of ROS in MF-induced SM hypoactivity, zebrafish embryos at 3 hpf were exposed to a 50-Hz MF at 200 μT with or without 200 μmol/L N-acetyl-L-cysteine (NAC) (a ROS scavenger) for 1 h or 24 h every day for 5 days. The ROS level in zebrafish larvae was analyzed by oxidation sensitive probe. Results found that MF exposure significantly increased ROS level in the head of zebrafishes (Figure 4A,B). Pretreatment with NAC, the MF-enhanced ROS in the head was descended to the normal level (Figure 4C,D), and MF-caused SM hypoactivity also could be restored in zebrafish (Figure 4E). It means that 50-Hz MF exposure-induced SM hypoactivity depended on ROS.

### 2.6. MF Regulated syn2a Expression via ROS

Since both syn2a and ROS were involved in MF-induced SM hypoactivity in zebrafish, in the present experiment, the relationship between syn2a and ROS was investigated. Zebrafish embryos at 3 hpf were exposed to a 50-Hz MF at 200 μT with 200 μmol/L NAC for 1 h or 24 h every day for 5 days. The syn2a expression in zebrafish larvae was analyzed by Western blot. Results showed that there was no significant difference in syn2a expression among all groups (Figure 5A), suggesting that pretreatment with NAC could effectively restore MF-inhibited syn2a expression. In addition, exposed zebrafish embryos at 3 hpf, which were transfected with plasmid vector or syn2a-overexpressed plasmid, to a 50-Hz MF at 200 μT for 1 h or 24 h every day for 5 days showed that syn2a overexpression did not affect MF-enhanced ROS level in zebrafish larvae (Figure 5B). All of the results indicated that MF-inhibited syn2a expression was mediated by ROS.

## 3. Discussion

Numerous studies confirmed that exposure to ELF-EMF could affect the function of nervous system, especially behavior, although the mechanism is still confusing [10]. SM is an important endpoint that reflects the developmental status and behavioral ability of zebrafish embryos/larvae. In the present study, the effects and underlying mechanisms of 50-Hz MF exposure on SM were investigated for the first time in zebrafish embryos/larvae. Results revealed that exposure of zebrafish embryos to a 50-Hz MF in two different modes with a series of intensities did not affect the basic developmental status, but could lead to early life-stage SM hypoactivity in a non-linear manner (at 200 μT). Since heat is an additive magnitude, an effect due to heat (i.e., a thermal effect) has to obey to a dose-response relationship, thus the non-linear trend of the detected MF effect on zebrafish larvae might be considered non-thermal. According to international standards (see ICNIRP 1998) [38], we assumed 50 Hz 100 μT (exposure limits) MF is the threshold for thermal/non-thermal exposure in humans, given the small size of the zebrafish larvae, a corresponding “man equivalent dose” (MEN) can be derived. The exposure of zebrafish, in our experiment (200 μT), resulting in a calculated current density of 190.99 nA/m^2^, was lower than the thermal threshold normalized for humans (about 1% MEN) (Giuliani L, 2023 personal communication), so this non-linear effect is non-thermal. The non-linear dose-response of MF, namely “window effect”, was first reported by Blackman et al. in 1980 [39]. It is defined as a biological response that occurs only under specific frequency, amplitude, intensity, and/or exposure mode criteria of MF. Outside these criteria, the biological response to MF is moderated or eliminated. The findings showed that 50-Hz MF-induced the “window effect” on SM hypoactivity in zebrafish was associated with intensity, but not exposure mode. Besides, it should be noted that, compared to the “frequency window effect” remaining relatively consistent (around 50/60 Hz), the “flux density/intensity window effect” discovered in studies for decades presented a significant heterogeneity. In the current experiment, we found that 50-Hz MF only at 200 μT could excite the inhibitory effect on SM, but other studies showed that 50/60 Hz MF, which affected the locomotor activity of animal, ranged in intensity from 95 μT to 4 mT [11,12,15,16,40,41]. Moreover, the effect of MF on SM was various. Some studies found that MF exposure inhibited locomotor activity [11,12,41], but others reported that MF promoted it [15,16,40]. Except of the diversity of animal species, the reason which caused the different effects on SM might be related to MF exposure duration. Available evidences indicated that the short-term (less than 4 days) exposure of MF was associated with SM inhibition [11,12,41], whereas long-term (more than two weeks) exposure would promote locomotor activity [15,16,40]. Additionally, the present study showed that exposing zebrafishes by MF in early life-stage could cause SM hypoactivity. Dimitrijević et al. also reported that MF exposure in early life-stage would decrease locomotor activity in *Drosophila subobscura* [13]. It suggested that the exposure stage in animal might influence the effect of MF on SM as well. Thus, the total effect of MF exposure on SM should be judged from multiple factors, at least including biological species, MF exposure duration and stage.

One of the main mechanisms studies on SM hypoactivity induced by environmental factors focus on the expression of neurodevelopment-related genes, such as α1-tubulin, gap43, gfap, mbp, shha, syn2a and so on [42,43,44,45,46,47,48,49]. The present study found that exposure to 50-Hz MF could down-regulate transcriptional levels of *α1-tubulin*, *gfap*, *mbp*, and *syn2a*, but only syn2a was down-regulated at the protein level, and confirmed for the first time that syn2a mediated the SM hypoactivity induced by 50-Hz MF in zebrafish. Syn2a is a subtype of the synapsins family, the first identified presynaptic neuronal phosphate protein and the most abundant protein on synaptic vesicles (SVs) involved in synaptogenesis and neuronal plasticity [50,51]. All isoforms of synapsins are involved in regulating γ-aminobutyric acid (GABA) release [50], but only syn2a regulates simultaneously glutamate (Glu) release by maintaining the stability of the SVs reserve pool of glutamatergic synapses [52]. Studies reported that ELF-EMF exposure could increase the ratio of Glu/GABA [53] and NMDA receptor activity [24,53,54,55] in the postsynaptic membrane. We could explore the expression of Glu and GABA in zebrafish larvae after MF exposure. As the major excitatory and inhibitory neurotransmitters in the central nervous system (CNS) respectively, the imbalance of Glu and GABA could influence locomotor behavior, even lead to neurological disorders such as ALS [56], epilepsy [57] and schizophrenia [58]. Interestingly, previous studies showed an increased risk of ALS in workers occupationally exposed to ELF-EMF [6,7,59], but the mechanism is still unclear. To some extent, the present findings suggest that MF exposure might induce ALS characterized by SM hypoactivity via disturbing syn2a-mediated the release of Glu and GABA. Regulating the balance of Glu/GABA by increasing the expression of syn2a might be a preventive measure for ALS occurred in workers occupationally exposed to ELF-EMF.

ROS is a definite target molecule for MF interacting with biological systems [37]. Hu et al. reported that ROS was involved in neurobehavioral disorders induced by nanoparticles in zebrafish larvae [42]. In the current study, results showed that ROS mediated MF-induced decrease in syn2a protein expression and caused SM hypoactivity. However, the exact mechanism is obscure. Our previous studies found that 50-Hz MF exposure could promote ROS production by increasing the intracellular Ca^2+^ level [60,61]. Coleman et al. reported that synapsin II actively delivered SVs from the reserve pool to the readily releasable pool in cooperation with Ca^2+^-dependent mechanisms [62], and Ca^2+^ could regulate Glu releasing from glutamatergic synapses [63]. All of these evidences indicated that MF-increased intracellular Ca^2+^ might participate in the inhibition of syn2a expression through regulation of ROS-mediated signaling, thereby affecting reserve pool stability at glutamatergic synapses, and accelerated SVs cycling to promote Glu release, ultimately lead to SM hypoactivity. But it requires further exploration. Additionally, related studies confirmed that syn2a is a substrate for some protein kinases activated by ROS, including cyclic adenosine monophosphate (cAMP)-dependent protein kinase (PKA) and extracellular signal-regulated kinase (Erk) [60,64,65,66], which are sensitive to ELF-EMF exposure [53,67,68]. Meanwhile, studies reported that both PKA [69,70] and Erk [71,72,73] were involved in locomotor behavioral regulation. Thus, it is possible that MF inhibits syn2a expression through these kinases, such as PKA and Erk, regulated by ROS, and causes SM hypoactivity finally. It provides a clue for the further investigation.

## 4. Materials and Methods

### 4.1. Chemicals and Antibodies

The main chemicals used in this study were N-acetyl-L-cysteine (NAC) (Adamas-Beta, Shanghai, China), Tricaine methanesulfonate (MedChemExpress, Shanghai, China) and E3 embryo medium (containing 5 mM NaCl, 0.17 mM KCl,0.33 mM CaCl_2_, 0.33 mM MgSO_4_, 0.01% methylene blue in water). The antibodies for Western blot analysis included rabbit anti-synapsin 1/2 antibody (Synaptic Systems, Goettingen, Germany), rabbit anti-alpha tubulin antibody (Abcam, Cambridge, UK), rabbit anti-glyceraldehyde-3-phosphate dehydrogenase (GAPDH) antibody (Bioss, Beijing, China), and horseradish peroxidases (HRPs)-conjugated secondary antibodies (Beyotime Biotech, Shanghai, China).

### 4.2. Zebrafish Husbandry, Spawning and Egg Collection

Adult zebrafish (wild-type AB strain, purchased from Zebrafish sub-platform of Zhejiang University Medical College Public Technology Platform) were cultured in tanks in a flow-through system (28 ± 1 °C,14 h:10 h light/dark cycle) according to the Institutional Animal Care and Use Committee protocols [74], and fed twice per day with freshly hatched live brine shrimp. Adult male and female zebrafishes were moved into the spawning tanks with a ratio of 1:1 in the afternoon of the day before spawning. The spawning was induced in the light after removing the baffle in the tank the next morning. Embryos were collected within 30 min, washed with E3 embryo medium, dead or unfertilized eggs were eliminated using a SMZ-168 stereo microscope (Nikon, Tokyo, Japan), incubated at 28 °C and marked as 0 hpf. Under microscopic observation, embryos at 3 hpf were randomly transferred into 5 mL E3 embryo medium in 6-well plates for the follow-up exposure experiments. The medium was changed daily and dead embryos were picked out. When experiment finished, 20 embryos or larvae from each well were collected and washed twice with standard water. All samples were stored at −80 °C for use.

This study was approved by the Experimental Animal Welfare Ethics Committee of Zhejiang University (Hangzhou, China) (Ethical batch No. ZJU20220209), and all of the experimental procedures were conducted in strict conformity with the guidelines for care and treatment of laboratory animals.

### 4.3. Magnetic Field Exposure System, Treatment of Zebrafish Embryos, and Observation of the Basic Developmental Parameters

The magnetic field exposure system (sXc-ELF) used in this study was purchased from the Foundation for Information Technologies in Society (IT’IS, Zurich, Switzerland). The detailed construction of the system was described in previous paper [75]. Briefly, it consisted of two exposure chambers in an incubator which could keep uniform temperature, and a set of control devices outside the incubator. One chamber was used for “MF exposure” and another for “sham exposure” (Figure 6A). The intensity of 50-Hz MF and the random selection of MF exposure chamber could be set by control devices for double blind experiments and there is no variation in the intensity of the magnetic field in every experiment (Figure 6B).

During MF exposure, the zebrafish embryos or larvae were put in the center of chambers, and the incubator is lighted. Temperature in the chambers was recorded in real time and kept at 28.0 ± 0.1 °C. The difference of temperature between the MF- and sham-exposure groups did not exceed 0.1 °C throughout the entire exposure period. The exposure system was turned on at least 2 h before an experiment for the conditions to stabilize. In the present study, zebrafish embryos were exposed to a 50-Hz MF with a series of intensities (100, 200, 400 and 800 μT, respectively) for 1 h or 24 h every day for 3 or 5 days.

After MF exposure, the larvae at 72 hpf or 120 hpf were collected for morphological observation, and the hatching rate at 72 hpf, mortality and malformation rate at 120 hpf were recorded using an optical microscope with camera (Nikon, Tokyo, Japan).

### 4.4. Overexpressing syn2a in Zebrafish

The synapsin IIa (syn2a) DNA sequences of zebrafish was downloaded from National Center for Biotechnology Information (NCBI). After synthesized, the DNA fragments were cloned into plasmid vector (pCS2^+^ plasmid) (Figure 7A) (Guannan Biological Technology Co., Hangzhou, China). An intact syn2a overexpression plasmid (pCS2^+^-CMVIE94-syn2a-EGFP plasmid) contains a green fluorescent protein (GFP) expression cassette composed of the Xenopus SP6 promoter for ubiquitous expression, the syn2a gene, the EGFP gene, and the SV40 polyA (Figure 7B). Plasmids were extracted using E.Z.N.A.^®^ Plasmid DNA Mini Kit I (Omega, NY, USA). The quality (OD260/280) and concentration of plasmids were measured by Nanodrop spectrophotometry (Thermo scientific, MA, USA). Total of 25 ng overexpression plasmid or plasmid vector was microinjected into the one-cell-stage zebrafish embryo before 1 hpf using the ASI pressure injection system under SMZ 745T stereo microscope (Nikon, Tokyo, Japan). Injected embryos were screened for green fluorescence at 24 hpf. The fluorescent images were observed and photographed by a SMZ18 microscope (Nikon, Tokyo, Japan).

### 4.5. Locomotion Analysis

At 24 hpf, six of live embryos in each group were selected randomly to count the times of spontaneous movement (SM, free swimming activity independent of any stimuli) within 1 min under microscope. Experiment was repeated at least three times.

At 120 hpf, a total of 12 healthy larvae randomly selected from each group were transferred into 96-well plates with one larva and 200 μL E3 embryo medium per well, which was placed in the Noldus DanioScope Observation Chamber (Noldus Industrial Co., Shanghai, China). After 10 min of acclimatization in the chamber, SM was monitored within 10 min with a Noldus EthoVision system (Noldus Industrial Co., Shanghai, China) at 28 °C. Besides, larva response to photoperiod stimulation was tracked during a cycle of two alternating periods of 10 min light (LED illumination) and 10 min dark (infrared illumination), after 10 min of acclimatization too. Using a Basler GenICam acA 130,060 g camera, the infrared movement traces were recorded at a rate of 25 frames per second and analyzed with the Noldus EthoVision XT 15 software. The total moving distance and average speed were determined during dark and light periods, respectively. Experiment was repeated at least three times.

### 4.6. Histological Examination

After exposure to a 50-Hz MF for 1 h or 24 h every day for 3 days, the larvae at 72 hpf were fixed with 4% (*w*/*v*) paraformaldehyde at 4 °C overnight, dehydrated in graded ethanol, cleared in xylene, embedded in paraffin, and then sectioned to 7 µm thickness. Subsequently, the sections were deparaffinized, rehydrated with graded ethanol, and stained with hematoxylin-eosin (HE) following the standard procedures. Finally, histopathological change of the brain was observed using a Ni-U microscope (Nikon, Tokyo, Japan).

### 4.7. ROS Assay

The ROS level in zebrafish larvae was analyzed by an oxidation sensitive probe, 2′,7′-dichlorofluorescein diacetate (DCFH-DA) (Beyotime Biotech, Shanghai, China) in vivo and in vitro. For ROS detection in vivo, the living larvae at 120 hpf which have been exposed to 50-Hz MF with or without 200 μmol/L NAC for 5 days were incubated with 5 μM DCFH-DA (final concentration in E3 embryo medium) for 1 h in the dark at 28 °C. After washing three times with E3 embryo medium, the larvae were anesthetized by MS-222 and observed using the fluorescence microscope (SMZ18, Nikon, Tokyo, Japan). The fluorescence intensity in the brain area was quantified by Image J software version 1.8.0.112 (NIH, Rockville, MD, USA).

For ROS assay in vitro, the heads of larvae which were treated in experiments were homogenized in ice and mixed with 600 μL 0.1 mol/L phosphate buffered saline (PBS, pH = 7.4). The supernatant was extracted by centrifugation at 12,500× *g* for 15 min at 4 °C, and the protein concentration was measured by a bicinchoninic-acid (BCA) protein assay kit (Beyotime Biotech, Shanghai, China). Each well of the 96-well blk/clr btm plate was added of 10 μL supernatant, 190 μL PBS and 0.1 μL 10 mM DCFH-DA respectively, and then incubated for 30 min at 37 °C in the dark. The intensity of fluorescence was detected by a SpectraMax^®^ iD5 multi-mode microplate reader (Molecular Devices, Sunnyvale, USA) at 480 nm excitation/530 nm emission and standardized to total protein levels. Experiment was repeated at least three times.

### 4.8. Quantitative Real-Time PCR (qRT-PCR) Analysis

Total RNA of zebrafish larva was isolated using RNAiso Plus (TAKARA BIO INC, Dalian, China). The quality (OD A260/A280) and concentration of RNA was measured by Nanodrop spectrophotometry (Thermo scientific, MA, USA). For each sample, cDNA was synthesized from 50 ng of total RNA using Prime Script^®^ RT reagent Kit with gDNA Eraser (TAKARA BIO Inc., Dalian, China) according to the manufacturer’s protocols [76]. Quantitative real-time PCR was performed by LightCycler^®^ 480 II System (Roche, Basel, Switzerland). The reaction system (total volume 10 μL) consisted of 5 μL of SYBR^®^ Green Realtime PCR Master Mix (TOYOBIO, Dalian, China), 1 μL of cDNA, 0.4 μL of forward and reverse primers (10 μM), and 3.2 μL of RNase-free H_2_O. Amplification was performed with the following cycling parameters: 95 °C for 30 s, 40 cycles of 95 °C for 5 s, 60 °C for 10 s, 72 °C for 15 s, followed by melt curve analysis to validate the specificity of the PCR amplicons. Aiming at evaluating the relative mRNA levels of genes, normalized to housekeeping gene β-actin, the comparative threshold cycle (2^−ΔΔCt^) method [77] was used. For each tested gene, the qRT-PCR reactions were performed on three technical replicates. The primer sequences of genes used in this study were presented in Table 3.

### 4.9. Western Blot Analysis

After treatment, zebrafishes were homogenized and lysed with 200 μL radioimmunoprecipitation assay (RIPA) buffer (Beyotime Biotech, Shanghai, China) containing 1 mM phenylmethylsulfonyl fluoride (PMSF) (Beyotime Biotech, Shanghai, China) and 4 mM cOmplete™-free protease inhibitor cocktail (Roche, Basel, Switzerland). Protein was extracted by centrifugation at 12,500× *g* for 15 min at 4 °C, and the concentration was measured using a BCA protein assay kit (Beyotime Biotech, Shanghai, China). Equal amounts of protein (30 µg/lane) were mixed with 5× loading buffer, resolved on 10% sodium dodecyl sulfate-polyacrylamide gel electrophoresis (SDS-PAGE) and transferred to nitrocellulose membranes (Whatman GmbH, Dassel, Germany). Then, the membranes were blocked with 5% bovine serum albumin (BSA) in tris-buffered saline (TBS) for 2 h, incubated overnight at 4 °C with rabbit anti-synapsin 1/2 antibody (1:1000), rabbit anti-alpha Tubulin antibody (1:2000) or rabbit anti-glyceraldehyde 3-phosphate dehydrogenase (GAPDH) antibody (1:2000), washed with TBST (TBS with 0.1% Tween-20) for 3 times (10 min per time) and incubated with goat anti-rabbit HRP (1:2000) for 1 h. Finally, the grey scale of protein bands was analyzed with BeyoECL on a Bio-Rad Chemiluminescence Imager (Bio-Rad, Hercules, CA, USA). Experiment was repeated at least three times.

### 4.10. Statistical Analysis

Data were presented as mean ± standard deviation (SD) from at least three independent experiments and analyzed using SPSS software version 26.0 (IBM, Armonk, NY, USA). After a normality test, the one-way analysis of variance (ANOVA) followed by least-significant difference or Dunnett post hoc comparison was performed to analyze the statistical differences among multiple groups. *p* < 0.05 was considered statistically significant.

## 5. Conclusions

The study demonstrated for the first time that 50-Hz MF exposure could induce zebrafish SM hypoactivity via ROS-inhibited syn2a expression in a nonlinear manner. The findings might provide new insights into the molecular mechanisms of ELF-EMF related neurodegenerative diseases such as ALS.

## Figures and Tables

**Figure 1 ijms-24-07576-f001:**
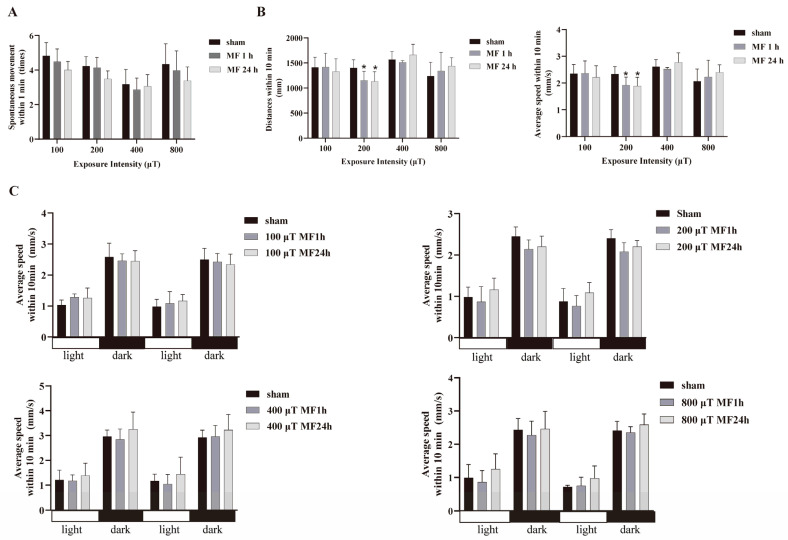
Effects of 50-Hz MF exposure on locomotor activity in zebrafish embryos/larvae. (**A**) MF exposure did not affect SM (spontaneous tail coiling) at 24 hpf within 1 min; (**B**) Exposure to MF with 200 μT decreased movement distance and average speed of SM at 120 hpf within 10 min; (**C**) MF exposure did not affect locomotor response to light-dark transition at 120 hpf. * *p* < 0.05, compared with sham.

**Figure 2 ijms-24-07576-f002:**
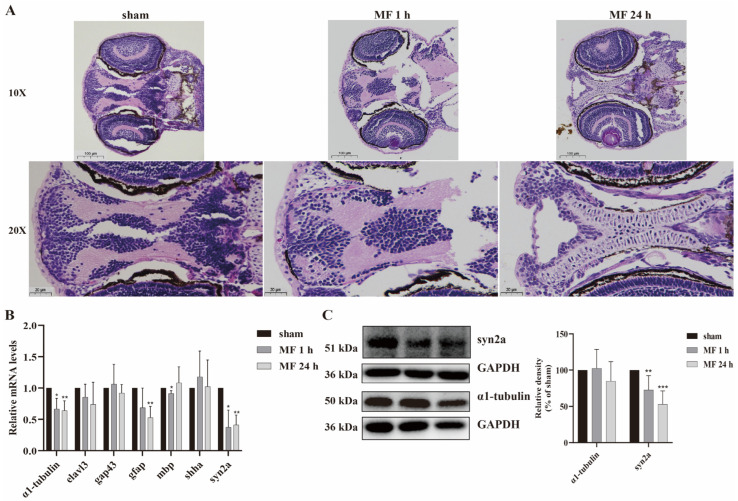
50-Hz MF exposure caused neurodevelopmental toxicity in zebrafish larvae. (**A**) MF exposure impacted the brain tissue morphology of zebrafish; (**B**) Exposure to MF decreased the transcription levels of neurodevelopment-related genes; (**C**) MF exposure inhibited significantly the expression level of syn2a. * *p* < 0.05, ** *p* < 0.01, *** *p* < 0.001, compared with sham.

**Figure 3 ijms-24-07576-f003:**
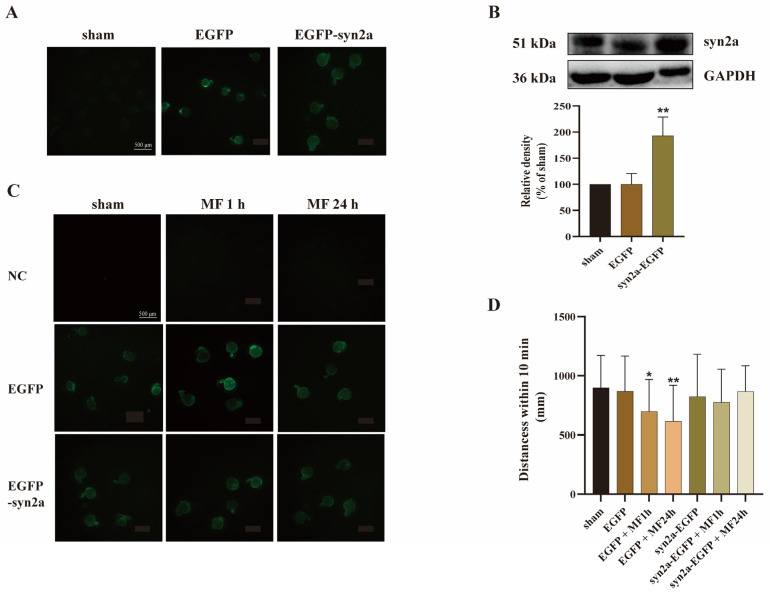
Overexpression of syn2a could restore MF-weakened SM in zebrafish. (**A**) EGFP could be expressed in zebrafish larvae which transfected with pCS2^+^ plasmid vector or pCS2^+^-syn2a-EGFP plasmid. Scale bar represents 500 μm; (**B**) Syn2a was overexpressed in zebrafish larvae transfected with pCS2^+^-syn2a-EGFP plasmid; (**C**) All of zebrafish embryos, which were transfected plasmid vector or syn2a overexpression plasmid and used in MF exposure experiment could express EGFP. Scale bar represents 500 μm; (**D**) Syn2a overexpression plasmid could restore MF-inhibited SM distances of zebrafish at 120 hpf. * *p* < 0.05, ** *p* < 0.01, compared with sham.

**Figure 4 ijms-24-07576-f004:**
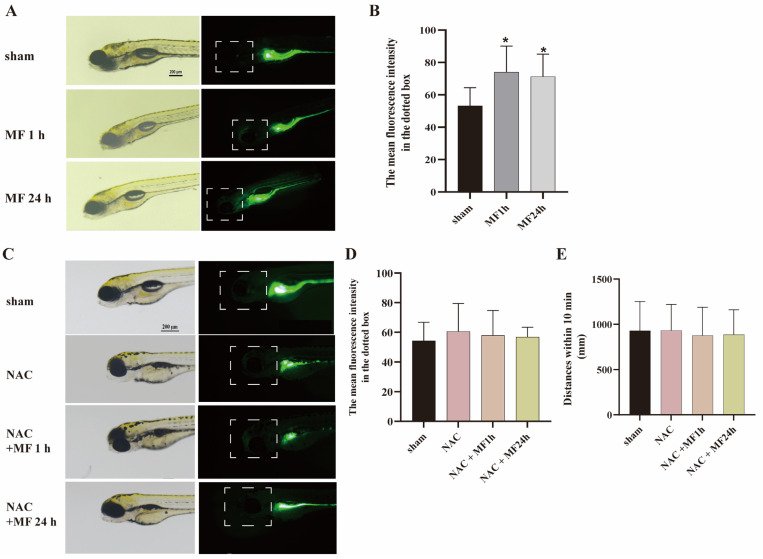
50-Hz MF exposure-induced SM hypoactivity depended on ROS. (**A**) MF exposure increased ROS level in the brain in zebrafish larvae. The left were photos in bright field light and the right were photos in blue excitation light under a fluorescence microscope. The dotted box was the location of the brain. Scale bar represents 200 μm; (**B**) The fluorescence intensity of ROS in the brain (dotted box) was increased after exposure to MF 1 h and 24 h every day for 5 days; (**C**) NAC pretreatment could inhibit MF-induced ROS in zebrafish larvae. The left were photos in bright field light and the right were photos in blue excitation light under a fluorescence microscope. The dotted box was the location of the brain. Scale bar represents 200 μm; (**D**) Pretreatment with NAC inhibited MF-increased ROS in the brain (dotted box) of zebrafish; (**E**) Pretreatment with NAC restored MF-inhibited SM distance of zebrafish at 120 hpf. * *p* < 0.05, compared with sham.

**Figure 5 ijms-24-07576-f005:**
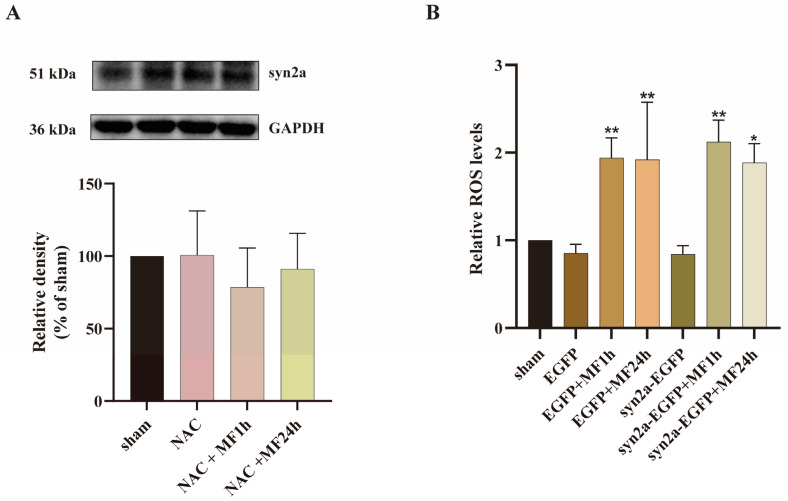
50-Hz MF inhibited syn2a expression via increasing ROS. (**A**) NAC pretreatment rescued MF-inhibited syn2a expression in zebrafish larvae; (**B**) Syn2a overexpression did not affect MF-enhanced ROS in zebrafish larvae. * *p* < 0.05, ** *p* < 0.01, compared with sham.

**Figure 6 ijms-24-07576-f006:**
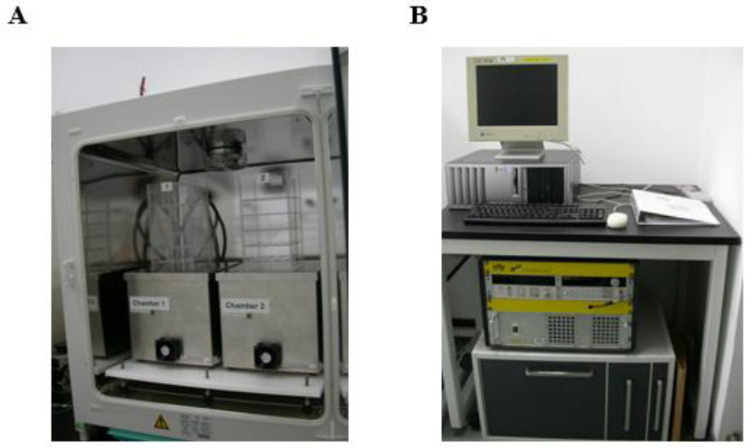
The 50-Hz magnetic field exposure system (sXc-ELF). It consists of two exposure chambers (**A**) and a set of control device (**B**).

**Figure 7 ijms-24-07576-f007:**
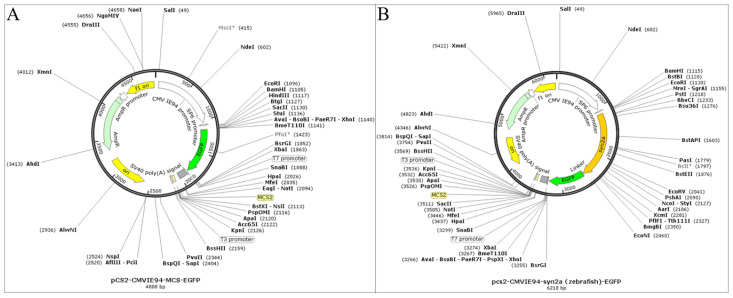
The construction of plasmid vector (**A**) and syn2a overexpression plasmid (pCS^2+^-CMVIE94-syn2a (zebrafish)-EGFP plasmid) (**B**). In the structural map of plasmids, some restriction sites (in gray) are marked with an asterisk in the upper right corner, indicating that there is more than a single restriction site. Restriction sites marked with an asterisk are generally not selected.

**Table 1 ijms-24-07576-t001:** Effects of 50-Hz MF exposure on embryonic development of zebrafish (*M* ± *SD*).

Developmental Index	100 μT			200 μT			400 μT			800 μT		
	Sham	MF1h	MF24h	Sham	MF1h	MF24h	Sham	MF1h	MF24h	Sham	MF1h	MF24h
Hatching rate (72 hpf)	95.40 ±2.17	96.17 ±1.36	92.90 ±1.57	84.64 ±4.12	89.52 ±4.28	94.11 ±2.91	95.99 ±2.07	94.71 ±2.24	94.38 ±2.41	94.50 ±2.01	93.30 ±1.59	92.80 ±1.47
Mortality (120 hpf)	9.72 ±0.83	8.82 ±2.36	7.10 ±1.57	4.84 ±3.30	5.99 ±2.30	8.72 ±2.30	7.63 ±1.76	9.63 ±0.74	11.01 ±1.19	7.60 ±3.10	6.50 ±0.95	9.44 ±1.25
Malformation rate (120 hpf)	5.57 ±0.74	5.39 ±0.33	6.67 ±0.53	6.43 ±1.50	5.89 ±2.10	7.41 ±1.70	6.51 ±0.73	5.88 ±1.53	5.04 ±1.02	5.78 ±0.94	6.33 ±0.23	6.82 ±0.91

**Table 2 ijms-24-07576-t002:** The spontaneous movement of 50-Hz MF exposure on zebrafish (*M* ± *SD*).

Index	100 μT			200 μT			400 μT			800 μT		
	Sham	MF1h	MF24h	Sham	MF1h	MF24h	Sham	MF1h	MF24h	Sham	MF1h	MF24h
Distances (mm)	1410.68 ±202.78	1417.06 ±274.08	1330.37 ±252.76	1471.30 ±163.95	1152.28 ±182.56 *	1132.64 ±195.43 *	1566.67 ±160.14	1512.73 ±33.98	1661.28 ±213.11	1237.34 ±275.94	1339.10 ±371.74	1437.65 ±169.00
Average speed (mm/s)	2.35 ±0.34	2.36 ±0.46	2.22 ±0.42	2.45 ±0.23	1.96 ±0.37 *	2.01 ±0.22 *	2.61 ±0.26	2.52 ±0.06	2.77 ±0.36	2.06 ±0.46	2.23 ±0.62	2.40 ±0.28

Exposure to MF with 200 μT decreased movement distance and average speed of SM at 120 hpf within 10 min. * *p* < 0.05, compared with sham.

**Table 3 ijms-24-07576-t003:** Primer sequences of genes for quantitative real-time PCR analysis.

Gene Name	Primer Sequence (5′-3′)	
	Forward	Reverse
*elavl3*	GTCAGAAAGACATGGAGCAGTTG	GAACCGAATGAAACCTACCCC
*gap43*	TGCTGCATCAGAAGAACTAA	CCTCCGGTTTGATTCCATC
*gfap*	GGATGCAGCCAATCGTAAT	TTCCAGGTCACAGGTCAG
*mbp*	AATCAGCAGGTTCTTCGGAGGAGA	AAGAAATGCACGACAGGGTTGACG
*shha*	AGACCGAGACTCCACGACGC	TGCAGTCACTGGTGCGAACG
*syn2a*	GTACCATGCCAGCATTTC	TGGTTCTCCACTTTCACCTT
*α1-tubulin*	AATCACCAATGCTTGCTTCGAGCC	TTCACGTCTTTGGGTACCACGTCA
*β-actin*	ATGGATGAGGAAATCGCTGCC	CTCCCTGATGTCTGGGTCGTC

## Data Availability

The data presented in this study are available upon reasonable request from the corresponding author.

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
