# Peer review of "50 Hz Magnetic Field Exposure Inhibited Spontaneous Movement of Zebrafish Larvae through ROS-Mediated syn2a Expression"

_ijms, 2023, doi:10.3390/ijms24087576_

Round 1

Reviewer 1 Report

Dear authors,

The science described in this paper is good, well written and innovative; however, the manuscript can be a little improved for readers. I have made some suggestions for the manuscript:

1. Abstract: line 20-24: The results can be clearer and more objective for a better understanding of the reader.

2. The introduction is adequate, well written and informative.

3. Results: The figures generated in the pdf have low resolution. Check the original figures for better resolution in the publication of the manuscript.

4. Table 1: the title of table 1 needs a better description of the experimental design to facilitate the reader's understanding.

5. Figure 1 was confusing, there are more graphics than mentioned in the text and unidentified graphics (a,b,c...).

6. Check figure 1B "diatances".

7. Figure 2A MF 1 hour: The histological section appears to be broken, check. Please insert scale bar. Are the histological sections evaluating the same height in the brain?

8. Line 145 – 146: Be more direct in explaining NAC and ROS.

9. Usually the statistical "p" is written in lowercase letter format; in some areas of the text it is capitalized.

10. Line 177 – 182: Be more direct in explaining MF window effect.

11. Line 212 – 216: This text is very speculative, as the study used fish larvae and in this text is comparing with diseases in human workers. I think that a correlation can be suggested, but in a less incisive way.

12. Materials and Methods: the model of the microscope and camera used in the experiments are interesting information to add in the manuscript.

13. The variation of the magnetic field in the exposure area must be described or make it clear that there is no variation in the intensity of the magnetic field.

14. Figure 7: the letters are very small and with low resolution, making it impossible to read.

15. Histological examination: line 327: Would the histological section be 7mm or 7µm thick?

16. ROS assay: line 340: insert the imagej software reference, including the functions or plugin used.

17. The conclusion is adequate.

I bring these issues to the author’s attention in the hope that the manuscript can be improved.

Reviewer 2 Report

In this study, a low frequency magnetic field exposition window for zebrafish embryos and larvae is identified (50 Hz, 200 muT), in which significant reduction of the spontaneous movement of 5 day old zebrafishes was triggered. To find a physiological reasons for this induced behavioral change, several tests have been carried out including transcriptome and protein expression via qRT-PCR and western blotting as well as histological tests. Having thus observed that the syn2a protein expression has been significantly down regulated in the critical exposition window, the authors transfect a syn2a overexpressed plasmid vector and showed that the movability reduction of the fishes could be mitigated in this way, proving that syn2a protein is relevant in this process. In addition, they showed that a scavenger for reactive oxygen species (ROS) also eliminated the hypoactivity of exposed zebrafishes, which leads to the hypothesis, that the magnetic effect is transmitted via ROS.

The authors present a strong result that should be published. The applied techniques are state of the art and mostly described in a sufficiently detailed way. The authors provide a complete discussion of their work and relate it to preceding work. However, before publishing, I would like to suggest that the authors review the text ones more considering the following points:

1.)   It would be interesting if the authors compared their findings on syn2a protein expression (with and without ROS scavenger) to samples exposed with a magnetic flux density outside the observed window for hypoactivity (e.g., 100 muT or 400 muT). The described phenomena must not occur in these cases if the proposed mechanism is really relevant for the hypoactivity of exposed zebrafishes.

2.)   It would be nice to have additionally a table like Table 1 for the results on the fishes’ mobility in terms of velocity and swum distance, although these data are already implicitly contained in the diagrams in Figure 1. It would also be nice, if the sample sizes would be also mentioned in this table. This would help to understand the outcome of the statistical analysis. 

3.)   What is the meaning of the integer number in Table 1? If this is not explained in the text, some information should be added.

4.)   Is the unit in Figure 1 A really “min”? It is written in the text, that the frequency of “spontaneous tail coiling” is depicted in Figure 1 A. A frequency should be measured in “1/min” (throughout the text “mins” should be replaced by “min” as abbreviation for minutes).

5.)   How should the scale in the diagrams in Figure 4 B, C be interpreted (it seems to be a percentage, but what does 100% mean)? Further, “fluorescence mean” does not refer to a physical quantity. Is maybe “mean intensity” of the fluorescence meant?

6.)   The term “light – dark transition” on page 3 is maybe misleading. As I understood, the usual light – dark cycle for zebrafishes has been implemented (which is important, as it has a strong influence on the later mobility of the fishes), and measurements have been carried out in both phases, but without a particular stimulus. The term “light – dark transition” suggests that a change of brightness has ben used as stimulus for fish activity, which seems not to be the case for the experiments carried out in this work. This should be clarified. In addition, the authors should add a sentence how they excluded all other known factors that could influence the mobility of zebrafishes.

7.)   In Section 4.3 some more information about the magnetic dose would be interesting, particularly uncertainty of the flux density, field direction relative to the probe, and special homogeneity (compared to the sample size). Can one assume that all parts of the fishes’ body are statistically exposed to the same field conditions? Is there a predominant direction of the magnetic flux? 

8.)   The discussion should be extended by a short remark, how further validation of the results could be achieved (other animal models to confirm the findings, other essays, etc.)

9.)   Although the structure of the article is very clear, the authors should add two or three lines in the end of the introduction to briefly outline the structure of the text.

10.)  The term “mediated” is differently used in throughout the text, which could be misleading. In the title, “ROS-mediated syn2a expression” is clear: it means that ROS trigger and control the process. “ROS-inhibited syn2a expression” as used in the conclusion is also clear. However, “… the possibility … that syn2a mediates [not: mediated] MF-induced SM hypoactivity” as written in the first sentence of Section 2.4 is misleading, because the lack of syn2a, which is inhibited or mediated by ROS, seems to trigger the hypoactivity. Maybe “… the possibility … that a reduced syn2a protein expression mediates MF-induced SM hypoactivity” is better here. The authors should check such formulations.

11.)  A strong argument in terms of validation is that qRT-PCR and western blotting yield the same results for 200 muT, because the methods are very different end support both a down-regulation of syn2a due to the magnetic exposure. This could be pointed out by the authors. Are the results also equivalent in the other cases (cf. item 1 in this list)? 

12.) The language throughout the whole text should be reviewed. Particularly the use of articles should be checked, singular and plural, choice of the tense (in most cases present tense is O.K.), and the use of active and passive voice. The line numbers in the following examples refer to the numbering in the first version of the article:

l. 31/32: low frequency electromagnetic fields … are generated

l. 41/42: could consequently show behavioral changes

l. 45/46: However, research on locomotor … has not reached a consensus.

l. 60: behavioral effects

l. 62: the zebrafish (Danio rerio) …

l. 63: the swimming behavior …

l. 83: spontaneous tail coiling of embryos

l. 86: of the embryos at 24 …

l. 87/88: no statistical significance compared with …

l. 88: However, exposure zebrafish embryos …

l. 92/93: in dark periods followed by a decreased swimming speed …

l. 94: no statistically significant deviation between different intensities 

l, 131: plasmid could mitigate MF-induced SM …

l. 139: level in zebrafish larvae …

l. 140: ROS level in the head of zebrafishes

l. 152: In addition, exposed zebrafishes embryos …

l. 163: revealed that exposure of zebrafish embryos …

l. 171: What is meant by “exposure mode” (wave form, field direction, etc.)? 

l. 178: Except of the …

l. 183: showed that exposing zebrafishes by MF in early …

l. 185/186: This work suggests that the exposure stage in animals might       influence … [maybe this sentence should be reformulated, because it does not give much information]

l. 187: … exposure on SM should be judged from multiple perspectives, at least …

l. 189: One of the main mechanisms studies … focus on is the expression of …

l. 202: in the central nervous system

l. 206/207: The present findings suggest that 

l. 211: with biological systems [37].

l. 221: thereby affecting reverse pool stability at

l. 223: But it requires further exploration.

l. 236: for Western blot analysis [western is also correct, but the spelling should be uniform]

l. 237: Göttingen [or Goettingen if ö is not available]

l. 250/251: medium, dead or unfertilized eggs were eliminated using a stereo

l. 270/271: for double blind experiments

l. 285/286: After synthesis, the DNA fragments were cloned into a plasmid vector 

l. 325: in zebrafish larvae was …

l. 329: for 1 h in the dark

l. 329/330: After washing three times with … 

Reviewer 3 Report

The authors presented a study under title of 50-Hz magnetic field exposure inhibited spontaneous movement of zebrafish larvae through ROS-mediated syn2a expression”.The applied methodology is sound. Also, the study looks good. Nonetheless, some issues need to be addressed.

Issues, weaknesses:

1.     The Abstracts must contain at least 150 words up to 250 words, and consist of 2-3 sentences as brief intro about the paper, 2-3 sentences to describe how the problem is solved, and 2-4 sentences showing the results of experiments/simulation ended with 1-2 sentences as short main conclusions of the work.

2.     The authors must have to include a performance comparison table in which they must have to compare this proposed work with minimum 10 previously reported similar types of works from 2018, 2019, 2020, 2021, 2022.

3.     Authors need to write their contribution clearly.

4.     The author's discussion of the latest research in the introduction is not sufficient, and it is recommended to add same new references.

5.     The Introduction section must explain the background of the problem and the urgency of the study, which can be proved by providing some previous researches and works, and also how to solve the problem in brief.

6.     The Results and Discussion section should give the analysis and explanation of all the result (Table and Figure). It is recommended to provide a comparison to a similar method from previous works and research.

7.     Comparison table should be added and proposed resonator should be compared quantitatively with other similar works.

8.     Abstract and conclusion parts should be revised carefully with quantitatively report with numbers and parameters improvement.

9.     Lack of novelty contributions of this manuscript.

10.  Tables 1 & 2 and Figures 1, 2, 3, 4, 5, 6 & 8 are not clear. Please provide clear version and correct the format issue. And please add more explanation on the legend so people can understand.

The revision has improved the quality of the manuscript, but the paper is still not organized well. Please recheck the format (manuscript, table and figures), writing style and grammar errors to meet the requirements.

Round 2

Reviewer 3 Report

Accept in current Condition. But the Tables and Figures are not clear 

Tables 1 & 2 and Figures 1, 2, 3, 4, 5, 6 & 8 are not clear. Please provide clear version and correct the format issue. 

Author Response

We have attached the original figures in zip format during the Round 2 of review. We will discuss with editor for how to improve it. Thanks for your suggestion.